# Scalable Deep Potentials as Implicit Hierarchical Semi-Separable Operators

## Abstract

1  Direct application of Transformer architectures in scientific domains poses
2  computational challenges, due to quadratic scaling in the number of inputs.
3  In this work, we propose an alternative method based on *hierarchical semi-*
4  *separable matrices* (HSS), a class of rank-structured operators with linear-
5  time evaluation algorithms. Through connections between linearized atten-
6  tion and HSS, we devise an implicit hierarchical parametrization strategy
7  that interpolates between linear and quadratic attention, achieving both sub-
8  quadratic scaling and high accuracy. We demonstrate the effectiveness of the
9  proposed approach on the approximation of potentials from computational
10  physics.

## 1  Introduction

12  *Many problems in computational physics require the evaluation of all pair-wise interactions in large*
13  *ensembles of particles* [1]. We consider learning scalar (*potential*) functions of the form

$$\Phi(x_\lambda) = \sum_{\mu=0}^{N-1} A(x_\lambda, x_\mu) v_\mu \tag{1.1}$$

14  where $x_\mu \in \mathbb{R}^d$ represent the generalized location (in a possibly high-dimensional abstract space)
15  of the particle, $A : \mathbb{R}^d \times \mathbb{R}^d \to \mathbb{R}$ is the associated *kernel* operator and $v_\mu \in \mathbb{R}$ is a physical feature
16  of each particle. Expressions of this type are pervasive and include electrical and gravitational
17  potentials, as well as other interaction potentials that play a pivotal role in determining forces and
18  influencing the dynamics of a system. Since the total number of particles in a system can grow large,
19  a model approximating $\Phi(x_\lambda)$ should offer an efficient evaluation algorithm in order to be utilized
20  in the inner loop of numerical solvers.

21  A rigorous analysis of Equation (1.1) reveals structural parallels with components of recent deep
22  learning architectures, most notably the self-attention mechanism intrinsic to the Transformer. In
23  this context, the kernel function $A(x_\lambda, x_\mu)$ delineated in our scalar potential formulation bears a
24  resemblance to the interaction computations inherent to the self-attention process. Specifically, the
25  linearized self-attention [2], [3] can be seen as a separable (low-rank) approximation of $A$

$$A(x_\lambda, x_\mu) = \sum_{\nu=1}^{p} \phi_\nu(x_\lambda) \psi_\nu(x_\mu) \tag{1.2}$$

26  where $\phi_\nu$ and $\psi_\nu$ are parametric functions (e.g. linear operators) $\mathbb{R}^d \to \mathbb{R}$, $\nu = 1, \ldots, p \ll N$.
27  The kernel function, which in the realm of physics might represent physical interactions between

particles, in the domain of deep learning encapsulates the interaction strengths between different tokens in a sequence. The separability property grants approximated kernels – and linear attention – a $\mathcal{O}(N)$ complexity to be evaluated.

Introducing the *softmax* normalization, which is customary in the Transformer's attention mechanism, results in an operator that deviates from the canonical form of (1.1) due to the normalization component. We write,

$$\mathsf{Att}(x, s(x)) = \sum_{\mu=0}^{N-1} \frac{1}{s_\lambda(x)} A(x_\lambda, x_\mu) v_\mu, \tag{1.3}$$

where $A(x_\lambda, x_\mu) = e^{\phi(x_\lambda)\psi(x_\mu)}$, $\quad s_\lambda(x) = \sum_{\gamma=0}^{N-1} e^{\phi(x_\lambda)\psi(x_\gamma)}$ and $v_\mu$ serves as the value in the Transformer. The appeal of the self-attention mechanism with softmax, despite its expressivity, is overshadowed by its computational constraints. Specifically, the kernel $A(x_\lambda, x_\mu) = e^{\phi(x_\lambda)\psi(x_\mu)}$ is inherently non-separable. Unlike separable kernels, where fast algorithms exist to exploit structure for efficient computation, non-separable kernels are bound to an $\mathcal{O}(N^2)$ complexity to evaluate the potential.

Existing approaches to approximate potentials rely on the application of stacks of self-attention operators, arranged in a Transformer architecture [4]. Other methods instead exploit locality assumptions and employ graph neural networks to reduce computational cost [5].

In this work, we aim to bridge the approximation capabilities of generic non-separable kernels such as self-attention with the fast evaluation of separable kernels. To do so, we devise a class of learnable kernels based on *hierarchical semi-separable* (HSS) matrices [6], [7]. Such matrices inherently support efficient matrix-vector multiplication due to their hierarchical, low-rank structure. HSS operators provide favourable rates of approximation for generic dense matrices while offering a tunable trade-off between computational overhead and rank of approximation, and further capture various other structured matrices arising in applications [8].

# 2 Hierarchical Semi-Separable Operators

The HSS representation of an operator $H \in \mathbb{R}^{N \times N}$ is obtained through a recursive row and column partitioning. A common partitioning strategy is to hierarchically bisect column and row indices up to a base level $L$, uniquely identifying $2^L$ blocks on the diagonal of $H$. Let such blocks be denoted as $D_m^L$ for $m = 1, 2, \ldots, 2^L$. We can then recursively compose increasingly larger blocks by bottom-up composition. Indeed, every HSS decomposition of this type can be paired with a binary tree, shown in Figure 2 for reference.

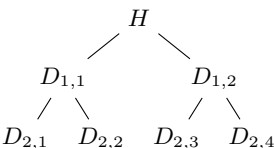

Figure 2.1: Complete binary associated with a $L = 2$ HSS decomposition of $H$

**Definition 2.1** (HSS matrix [7]). *A matrix $H$ is said to be hierarchically semi-separable (HSS) if there exist matrices $D^\ell \in \mathbb{R}^{N/2^\ell \times N/2^\ell}$, $U^\ell \in \mathbb{R}^{N/2^\ell \times r}$, $V^\ell \in \mathbb{R}^{r \times N/2^\ell}$, $R^\ell \in \mathbb{R}^{r \times r}$, $W^\ell \in \mathbb{R}^{r \times r}$, $B^\ell \in \mathbb{R}^{r \times r}$ that satisfy the following recursion:*

$$\begin{aligned}
D_m^{\ell-1} &= \begin{pmatrix} D_{2m-1}^\ell & U_{2m-1}^\ell B_{2m-1}^\ell V_{2m}^{\ell\top} \\ U_{\ell;2m} B_{\ell;2m} V_{\ell;2m-1}^\top & D_{\ell;2m} \end{pmatrix}, \\
U_m^{\ell-1} &= \begin{pmatrix} U_{2m-1}^\ell R_{2m-1}^\ell \\ U_{2m}^\ell R_{2m}^\ell \end{pmatrix}, \quad V_m^{\ell-1} = \begin{pmatrix} V_{2m-1}^\ell W_{2m-1}^\ell \\ V_{2m}^\ell W_{2m}^\ell \end{pmatrix}, \\
m &= 1, 2, \ldots, 2^{\ell-1}, \quad \ell = 0, 2, \ldots, L.
\end{aligned} \tag{2.1}$$

*and the condition $D_1^0 = H$.*

> **Note:** We are interested in hierarchical matrices with fast evaluation algorithms. Thus, we seek factors $UBV^\top$ where either the rank $r$ of $B$, is sufficiently small or the decomposition admits a fast evaluation algorithm itself. In example, let $U$ and $V$ be diagonal matrices, and further let $B$ be Toeplitz. Then, $Bu$ can be evaluated in $\mathcal{O}(N \log N)$ via a Fast Fourier Transform [9]. When the off-diagonal terms are not low-rank, we refer to this class of operators as *pseudo-HSS*, due to their hierarchical structure.

HSS matrices have found use in deep learning architectures, as a way to replace generic dense weight matrices [10], [11]. Instead, we seek to develop an efficient *implicit* class of learnable HSS kernels.

# 3  Learning via Implicit HSS

The self-attention kernel is a canonical example of an *implicit* operator. Implicit operators are effective primitives for architecture design, as they decouple parameter counts from some critical input dimensions[1].

We can therefore find a link between attention and HSS through linear attention, which is low-rank and hence separable:

**Theorem 3.1.** *Linear self-attention is HSS.*

A sketch of the proof is provided in Appendix A. Other subquadratic implicit operators commonly used as attention replacements can similarly be shown to satisfy (2.1).

**Corollary 3.1** (Efficient implicit operators are HSS). *Sparse attention, gated convolutions and recurrences (Hyena [12], H3 [13], S4 [14]) are pseudo-HSS.*

Dense attention, however, is not separable, and thus is not HSS. This is due to the nonlinearity introduced via softmax. A hierarchical pseudo-HSS approximation, however, can be given as the following:

**Definition 3.1** (IHSS). *Let $H(x)$ be an implicit operator defined via the recurrence*

$$
D_m^{\ell-1} = \begin{pmatrix} D_{2m-1}^\ell & \mathsf{S}(U_{2m-1}^\ell B_{2m-1}^\ell V_{2m}^{\ell\top}, s_{2m-1}^\ell + \beta_{2m-1}^\ell) \\ \mathsf{S}(U_{\ell;2m} B_{\ell;2m} V_{\ell;2m-1}^\top, s_{2m}^\ell + \beta_{2m}^\ell) & D_{2m}^\ell \end{pmatrix},
$$
$$
U_m^{\ell-1} = \begin{pmatrix} q_{2m-1}^\ell \\ q_{2m}^\ell \end{pmatrix} \quad V_m^{\ell-1} = \begin{pmatrix} k_{2m-1}^\ell \\ k_{2m}^\ell \end{pmatrix}, \quad \beta_m^{\ell-1} = \begin{pmatrix} \beta_{2m-1}^\ell \\ \beta_{2m}^\ell \end{pmatrix} + \begin{pmatrix} s_{2m-1}^\ell \\ s_{2m}^\ell \end{pmatrix}
$$
$$
m = 1, 2, \ldots, 2^{\ell-1}, \quad \ell = 0, 2, \ldots, L.
$$

(3.1)

*with leaf nodes $D_m^L = \mathsf{Att}(u_m^L)$, where we denote with $\mathsf{S}(U, s)$ elementwise exponentiation of the matrix $U$, and row-wise division by elements of the state $s$. Then, $H(x)$ is a* IHSS.

The main idea behind the above is to leverage the tree structure of a pseudo-HSS decomposition to obtain an approximation of attention with fewer operations.

**Lemma 3.1** (Cost of IHSS). *Evaluating partial results $D_m^\ell(u_m^\ell)u_m^\ell$ for $m = 1, 2, \ldots, 2^{\ell-1}$ of* IHSS *requires $\mathcal{O}(2^{2n-\ell}d)$ arithmetic operations, with $n = \log_2 N$*

Note that if $\ell = \log_2 N = n$, the asymptotic complexity is linear in the number of particles. This suggests a path forward: we can hybridize IHSS, performing $\log_2 N$ levels of the approximation (3.1), then complete the bottom-up recursion with a different, linear-time IHSS recursion (e.g., linear attention) for levels $L + 1 - \log_2 N$ levels.

## 3.1  Additional properties of IHSS

**Directional approximation**  The IHSS is a *directional* approximation of softmax attention. Directionality is a consequence of the state $s$ in the softmax function, which couples elements across

---

[1]In the attention example, parameter counts are independent of $N$, instead scaling as $\mathcal{O}(d^2)$.

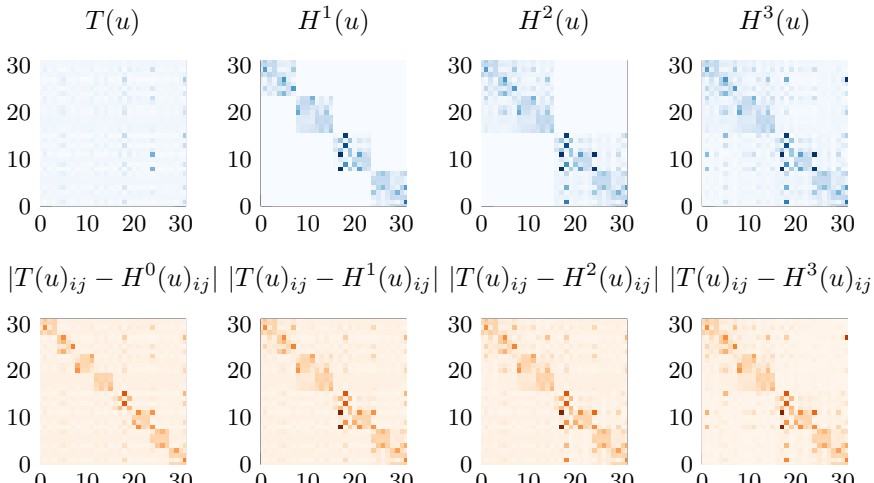

Figure 3.1: **[Top]:** Attention $T(u)$ and `IHSS` with $L = 3$. **[Bottom]:** Approximation error. The average error is minimum at the last level, in off-diagonal blocks.

blocks. Traversing the HSS tree in a bottom-up fashion, the approximation error between attention and `IHSS` on the off-diagonal entries decays, as shown in Figure 3.1. One implication of this property is that `IHSS` is a more accurate approximation of attention on long interactions.

A similar argument can be used for hierarchical decompositions of implicit operators with other coupling operations, by augmenting the recurrence with additional states[2].

**Local permutation equivariance**   `IHSS` is equivariant to structured permutations that preserve some block membership. In particular,

$$H(Pu) = PH(u)$$

will hold if $Pu$ shuffles elements inside any linear attention levels of a hybrid `IHSS` or if the permutation shuffles the elements of leaf blocks $D_m^L$. Equivariance can be a desirable property in the task of approximating potentials in computational physics.

## 4   Numerical Experiments

We investigate how accurately different implicit operators can approximate example potentials with different characteristics. Denote with $d_{\lambda,\mu}$ the distance between $d(x_\lambda, x_\mu)$ We consider:

- Coulomb-like potentials:

$$A(d_{\lambda,\mu}) = \frac{1}{d_{\lambda,\mu}}$$

- Lennard-Jones potential:

$$A(d_{\lambda,\mu}) = \frac{1}{d_{\lambda,\mu}}^6 - \frac{1}{d_{\lambda,\mu}}^{12}$$

- Morse potential

$$A(d_{\lambda,\mu}) = (1 - e^{-d_{\lambda,\mu}})^2.$$

**Protocol**   We prepare a dataset of $32k$ samples on a one-dimensional domains. Each sample in the dataset contains a $256$ or $8192$ particles, with positions sampled from an isotropic Gaussian distribution. Figure 4.1 shows the scalar potentials on the sorted particle positions. We train single

---

[2]In (3.1), $\beta \in \mathbb{R}^N$ acts as an accumulator for the softmax normalizing factors.

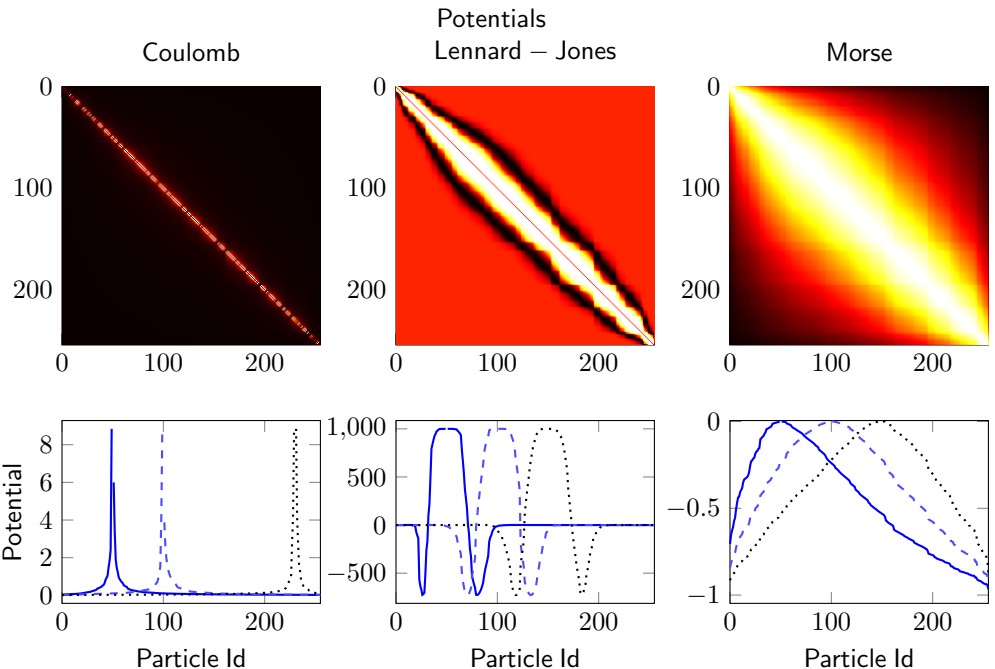

Figure 4.1: Target potential functions considered for the approximation task.

| Implicit Operator | Coulomb | | Lennard-Jones | | Morse | |
|---|---|---|---|---|---|---|
| | 256 | 8192 | 256 | 8192 | 256 | 8192 |
| Attention | 0.211 | 0.128 | 0.358 | 0.327 | 0.104 | 0.074 |
| Linear Attention | 0.207 | 0.112 | 0.391 | 0.321 | **0.081** | **0.063** |
| Hyena | 0.189 | **0.107** | 0.298 | **0.264** | 0.083 | 0.067 |
| IHSS | **0.172** | 0.124 | **0.294** | 0.278 | **0.081** | 0.066 |

Table 4.1: Validation loss of different methods.

layer models comprised on an implicit operator in the class of self-attention [15], linear attention [2], Hyena [12] and IHSS with $d = 32$. We apply RBF positional embeddings to the particle position, following [4]. All models are optimized with the Adam optimizer, learning rate $10^{-3}$, 1000 epochs, cosine scheduler down to $10^{-4}$. The loss function is normalized mean-squared error. Table 4 reports validation loss in different experimental setups. IHSS is competitive with other implicit operators, consistently outperforming self-attention. Note that all operators have less than 2000 learnable parameters, which is $3\%$ of the $65536$ values required to represent the potential when $N = 256$.

# 5 Conclusion

In this work, we introduced the IHSS, an implicit parametrization for hierarchical kernels that interpolates between linear and quadratic attention, achieving both subquadratic scaling and high accuracy. Through numerical experiments, we demonstrated its competitive performance against existing kernels such as self-attention [15].

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

## A Proofs

**Lemma A.1** (Linear Attention is HSS). *Linear self-attention $A(x)v = q(x)k(x)^T v(x)$ is HSS.*

*Proof.* To show that linear self-attention $A(x)v = q(x)k(x)^T v$ is HSS, we need verify that the matrices $D_m^\ell$, $U_m^\ell$, and $V_m^\ell$ can be constructed in a way that satisfies the recursion formula given by HSS.

For the base case of $\ell = L$, let $D_m^L = q_m^L k_m^{L\top} = A(x_m^L)$.

At each lower level $\ell$, we construct $D_m^{\ell-1}$, $U_m^{\ell-1}$, and $V_m^{\ell-1}$ from the level $\ell$ as

$$D_m^{\ell-1} = \begin{pmatrix} D_{2m-1}^\ell & q_{2m-1}^\ell k_{2m}^{\ell\top} \\ q_{2m}^\ell k_{2m-1}^{\ell\top} & D^{\ell 2m} \end{pmatrix}, \qquad m = 1, 2, \ldots, 2^{\ell-1}$$

$$U_m^{\ell-1} = \begin{pmatrix} q_{2m-1}^\ell \\ q_{2m}^\ell \end{pmatrix} \quad V_m^{\ell-1} = \begin{pmatrix} k_{2m-1}^\ell \\ k_{2m}^\ell \end{pmatrix}.$$

The induction step is

$$D_m^{\ell-1} = \begin{pmatrix} q_{2m-1}^\ell k_{2m-1}^{\ell\top} & q_{2m-1}^\ell k_{2m}^{\ell\top} \\ q_{2m}^\ell k_{2m-1}^{\ell\top} & q_{2m}^\ell k_{2m}^{\ell\top} \end{pmatrix} = q_m^{\ell-1} k_m^{\ell-1\top}.$$

This implies the HSS recurrence will terminate with the linear attention matrix at root level $\ell = 0$.

$\square$

