# OpenReview forum: "Scalable Deep Potentials as Implicit Hierarchical Semi-Separable Operators"
_NeurIPS.cc/2023/Workshop/AI4Science — NeurIPS2023-AI4Science Poster_

### Official Review · Reviewer_kbGC · 2023-10-21
**Review of 'Scalable Deep Potentials as Implicit Hierarchical Semi-Separable Operators'**

**Rating:** 7
**Confidence:** 3

**Review:**

This paper proposes an interesting approach, Implicit HSS, to realize scalable neural potentials. This approach has been evaluated on Coulomb-like, L-J and Morse potentials and show good performance for datasets with varying particle numbers.

Pros: the authors did a good job on the theoretical derivation of their framework, and discussed its relations between linear and non-linear attention mechanisms.

Cons: the experimentation results part is a bit weak, with a few comments (1) Can authors provide more technical details about Hyena, and how would that compare with the other two attention mechanisms? Why does Hyena give better accuracy for two large-scale cases? (2) The proposed IHSS method achieved good accuracy for all small-scale systems, but why do we not see improved accuracy for the 8k-sized particle systems? This seems to be a bit contradictory to authors' claim on scalability. (3) what part of the results indicate the better scalability of IHSS than baseline models?

Also, I would like to see some brief discussions on IHSS's applicability to more complex particle systems than L-J or coulomb etc. perhaps as a vision for future work in the conclusion part.

---

### Official Review · Reviewer_EnxG · 2023-10-25
**Interesting work geared towards the scientific computing community**

**Rating:** 7
**Confidence:** 2

**Review:**

I thought this paper is interesting, and quite mathematically involved. It uses the idea of implicit operators to reduce the computation of potentials associated with pair-wise interactions. The complexity of these problems are usually quadratic in size, just like the cost of running self-attention neural networks. However, HSS approximation makes this problem sub-quadratic by approximating pair-wise interaction using hierarchical semi-separable operators.

**Pros:**
This work is fairly clearly written. I am not an expert in computational physics, but was able to grasp the gist.

**Cons/room for improvement**
Lack of bench marking. How does the proposed method compare with other methods with similar compression ratios, such as GNN, which is mentioned?

Physical intuition. In real-world computations, all approximations hinges on assumptions about the physical problem. For example, GNN with near-neighbor pooling assumes dominating local interaction between particles. What are the physical assumptions that make HSS approximation valid? Where does the low-rankness come from?

---

### Meta-Review · Area_Chair_edi6 · 2023-10-26

**Recommendation:** Accept (Poster)
**Confidence:** 2

**Metareview:**

All reviewers agree that the paper is well fit to the workshop scope and provide a valuable contribution to the community. Please try to integrate reviewers' comments in the later revision.